# Molecular Determinants of the Early Life Immune Response to COVID-19 Infection and Immunization

**DOI:** 10.3390/vaccines11030509

**Published:** 2023-02-22

**Authors:** Elisabeth M. S. Beijnen, Oludare A. Odumade, Simon D. van Haren

**Affiliations:** 1Precision Vaccines Program, Division of Infectious Diseases, Boston Children’s Hospital, Boston, MA 02115, USA; 2Harvard Medical School, Boston, MA 02115, USA; 3Department of Pediatrics, Division of Medicine Critical Care, Boston Children’s Hospital, Boston, MA 02115, USA

**Keywords:** COVID-19, children, vaccine, infection

## Abstract

Clinical manifestations from primary COVID infection in children are generally less severe as compared to adults, and severe pediatric cases occur predominantly in children with underlying medical conditions. However, despite the lower incidence of disease severity, the burden of COVID-19 in children is not negligible. Throughout the course of the pandemic, the case incidence in children has substantially increased, with estimated cumulative rates of SARS-CoV-2 infection and COVID-19 symptomatic illness in children comparable to those in adults. Vaccination is a key approach to enhance immunogenicity and protection against SARS-CoV-2. Although the immune system of children is functionally distinct from that of other age groups, vaccine development specific for the pediatric population has mostly been limited to dose-titration of formulations that were developed primarily for adults. In this review, we summarize the literature pertaining to age-specific differences in COVID-19 pathogenesis and clinical manifestation. In addition, we review molecular distinctions in how the early life immune system responds to infection and vaccination. Finally, we discuss recent advances in development of pediatric COVID-19 vaccines and provide future directions for basic and translational research in this area.

## 1. Introduction

Severe Acute Respiratory Syndrome Coronavirus-2 (SARS-CoV-2) is an enveloped single-stranded, positive-sense ribonucleic acid (ssRNA+) coronavirus that belongs to the family of Coronaviridae [1]. The coronavirus disease (COVID-19) manifests itself as mild–moderate symptoms, such as coughing, muscle fatigue and fever to severe respiratory failure that requires ICU care and ventilation [2]. In children, however, clinical manifestations from primary COVID infection mostly remain limited. Severe pediatric cases occur predominantly in children with underlying medical conditions such as cardiac and circulatory congenital anomalies, type I diabetes [3] and asthma [4]. Despite the lower incidence of disease severity, the burden of COVID-19 in children is not negligible. With a growing proportion of older Americans vaccinated, and the rise of delta and omicron variants, case incidence in children has substantially increased with estimated cumulative rates of SARS-CoV-2 infection and COVID-19 symptomatic illness in children aged 5–17 years comparable to those in adults aged 18–49 and higher than those in adults aged > 50 years [5,6]. Per the U.S. Centers for Disease Control (CDC), there have been >800 pediatric COVID deaths in the U.S. alone. The American Academy of Pediatrics (AAP) indicates that, as of 25 August 2022, 14,448,622 total child COVID-19 cases were reported, and children represented 18.4% of all reported cases (https://www.aap.org/en/pages/2019-novel-coronavirus-COVID-19-infections/children-and-COVID-19-state-level-data-report, accessed on 29 November 2022). Children, and especially young infants, can spread SARS-CoV-2 to older individuals including parents, grandparents, and teachers [7]. Although SARS-CoV-2 virus shedding varies among individuals [8], infected children can shed SARS-CoV-2 virus with nasopharyngeal viral loads comparable to or higher than those in adults [9,10]. Emerging literature on pediatric COVID-19 is focused on finding an explanation for the lower severity of infection observed in this population [11,12,13]. While a definitive answer to this query is unknown, age-related differences in immune system function may explain the higher susceptibility to serious events in adults.

The immune response to infections or vaccinations varies with age, especially during childhood. Functional distinctions in immune cell behavior have been demonstrated by our group and others, reflecting unique age-dependent challenges including gestation, the neonatal phase, and infancy [14,15,16,17,18,19,20,21,22]. Activation of the innate immune system through Pattern-recognition receptors (PRRs), such as Toll-like Receptors (TLRs), differs markedly with age. The magnitude of cytokine production in response to most TLR ligands is impaired in newborns and infants [18,23], with the exception of TLR7/8 ligands [24,25].

A key feature of the distinct innate responses in early life is that the production of cytokines is polarized towards high production of T-helper-2 promoting or anti-inflammatory cytokine such as IL-6 and IL-10 [19,26], and is impaired in the production of key T-helper 1 promoting cytokines such as TNF and IL-12p70 [15,27]. Similarly, human and murine neonatal macrophage production of immunosuppressive IL-27, a heterodimeric cytokine of the IL-12 family, peaks during infancy [28]. Because IL-12p70 is predominantly derived from circulating dendritic cells, the slow maturation of IL-12p70 synthetic capacity in the childhood years can be attributed to reduced numbers and/or functionality of these dendritic cells [29]. Changes during childhood in adaptive immune responses include a reduced capacity in antigen internalization, processing, MHC II presentation, and subsequent stimulation of antigen-specific CD4+ T cell responses [30,31]. The resulting impairment in early life to promote T-helper 1 immunity to novel pathogens can prime the immune system of children to the preferential induction of a tolerogenic or sometimes detrimental T-helper 2 response [29,32,33]. While these intrinsic features of the developing immune system may have proven disadvantageous in the context of other respiratory viruses such as RSV [34,35,36,37,38], it has been postulated that in the context of SARS-CoV-2 this may contribute to reduced disease burden in this age group [11]. In addition, investigation into the age-specific differences in pathology and immune response to infection could contribute to improved future vaccine formulations tailored for children. Therefore, this article aims to systematically review age-dependent molecular determinants that could inform future COVID-19 pediatric vaccine development. The search criteria illustrated in Figure 1 resulted in a collection of 69 original research manuscripts that were included into the narrative synthesis of this review and were supplemented with references to additional work for contextualization. The search criteria, as well as the review itself, are centered around two main questions. The first was, can we identify changes with age in molecular determinants that contribute to susceptibility and pathogenesis? The second question was, can we identify changes with age in immune cell functionality that contribute to efficacy and durability of vaccination against SARS-CoV-2? Understanding the molecular distinctions in SARS-CoV-2 pathogenesis and vaccine responses between children and adults could directly inform future development of vaccine formulations tailored for early life.

## 2. Infection

### 2.1. Pathogenesis

The first step in the pathogenesis of COVID-19 after invasion is receptor recognition. SARS-CoV-2 is covered by many transmembrane Spike (S) glycoproteins. The S protein consists of two functional subunits, S1 and S2, which are responsible for receptor binding and membrane fusion, respectively [39]. The S1 subunit contains a receptor-binding domain (RBD) that specifically recognizes angiotensin-converting enzyme (ACE2) as its cellular receptor. The ACE2 receptor is widespread and distributed throughout human epithelial tissues, such as the nasal and oral mucosa, lungs, and heart [40]. It is argued that SARS-CoV-2 can enter the target cell via the cell surface or through endocytosis. Subsequently, the S-protein is cleaved by either the membrane-anchored serine protease TMPRSS2 or by lysosomal cathepsins [41,42]. It has been suggested that the cell surface receptor Neuropilin-1 (Nrp1) facilitates an earlier separation of the S-protein and therefore may influence virus infectivity [43]. Proteolytic cleavage induces conformational changes within the S2 unit leading to viral and host cell membrane fusion [44]. Interestingly, it has been described that the omicron variant favors the endosomal pathway over TMPRSS2-mediated entry [45]. This entry preference is probably due to a mutational reorganization of the S1-S2 cleavage site of its spike protein causing impaired fusion capability [46]. Other mutations in the RBD of the omicron strain are believed to be associated with decreased ACE2 binding affinity [47]. These observations may explain the lower pathogenicity observed in the omicron variant compared to the delta variant. Upon cell entry, viral contents are released into the target cell following viral RNA replication, transcription, and translation. Further processing of the S protein is conducted by furin prior to the release of new viruses into the extracellular space [41]. It should be noted that there are multiple proteases involved in the activation of the S protein that operate at distinct sites. The exact molecular entry and replication mechanism of SARS-CoV-2 is a complex process and therefore remains to be fully elucidated. 

### 2.2. Clinical Presentation (Children vs. Adults)

Global vaccine strategy efforts have led to a significant reduction of symptoms in COVID-19-infected individuals [48]. Clinical features rely inter alia on vaccination status, comorbidities, and other determinants of health such as residential area and age. Among the fully vaccinated, sore throat, cough, and fever are still predominant signs of delta-variant-infected individuals [48]. Interestingly, asymptomatic cases are frequently observed among the elderly (60 years or older) who are fully vaccinated [49]. In contrast, unvaccinated delta variant patients show a varying set of clinical features such as shortness of breath, gastrointestinal problems, olfactory impairment, and headache [50].

The omicron strain was named the fifth variant of concern by the World Health Organization [51]. From a clinical perspective, omicron infections are associated with a significant lower risk of severe disease when compared with delta variant infections [52]. This observation supports the difference in pathogenesis between these strains as described earlier. 

In its most severe form, an infection with COVID-19 could lead to an unfettered release of pro-inflammatory cytokines. This so-called cytokine storm is mainly observed in COVID-19 patients with severe infection [53]. In these patients, the hyperactive immune activation may progress to acute respiratory stress syndrome (ARDS) and multiple organ dysfunctions [54] and eventually death. 

The course of infection in children is usually asymptomatic or mild. Pediatric symptoms may vary and depend on different factors. In most cases, however, clinical features include classic flu-like symptoms such as pyrexia, sore throat, nasal congestion, and cough [12]. Besides the involvement of the upper airways, other organs such as the gastrointestinal system and the central nervous system could be affected. While most children experience an uneventful course, a small percentage develop multisystem inflammatory syndrome (MIS-C). This potentially life-threatening disease is characterized by an excessive immune reaction and typically manifests itself two to six weeks after COVID-19 infection [55]. In these patients, main symptoms include fever, and gastrointestinal and abdominal pain. Cardiac manifestations such as myocarditis and arrhythmias are also common [56]. Notably, respiratory tract involvement is not prevalent in MIS-C patients, which may help distinguish MIS-C from severe COVID-19 illness. Due to its clinical similarities, MIS-C was initially called Kawasaki-like disease. However, emerging literature points out interesting differences between these two conditions. For example, immunological research showed that children with MIS-C harbor a different cytokine profile in contrast to children who suffer from Kawasaki disease, suggesting a difference in underlying immunopathology [57,58]. In support of the foregoing, autoantibody responses involved in Kawasaki syndrome show clear differences when compared to autoantibody profiles in MIS-C patients [59]. It should be noted, however, that sample sizes included in these studies were small and further research should determine whether the previous findings are valid.

### 2.3. Differences in Entry Mechanisms between Children and Adults 

As stated in the previous chapter, SARS-CoV-2 may enter the target through the endosomal pathway or via TMPRSS2. Expression of TMPRSS2 is age-determined (Table 1, Figure 2), and alternative entry mechanisms in children could therefore be dominant. It should be noted that cells expressing both ACE2 and TMPRSS2 in infants are scarce [60], which could offer a protective mechanism against strains that prefer TMPRSS2-mediated entry. Another study also found lower TMPRSS2 expression in airway cells from children (age 0–17), but noted a higher expression of ACE2 [61].

Table 1 shows that TNFα-converting enzyme (TACE) expression is lower in children than in adults. TACE causes membrane-bound ACE2 to ‘shed’ from endothelial cells, generating its soluble and enzymatically active form. While the impact of soluble ACE2 (sACE2) on viral entry remains poorly understood, it has been suggested that sACE2 facilitates viral entry and thus promotes disease severity [62,63]. However, contradictory findings show that high sACE2 levels offer protection against SARS-CoV-2 [64]. In support of this hypothesis, newborns and infants exhibit higher sACE2 levels than adults [65]. Furthermore, studies have demonstrated that pro-inflammatory cytokines such as Il-1β and TNFα enhance ACE2 shedding. The levels of these cytokines are higher in children infected with SARS-CoV-2 compared to their infected adult counterparts (Table 2, Figure 2). Therefore, these molecules may enhance protectivity by providing greater resistance to SARS-CoV-2 in the pediatric population, if high sACE2 levels are in fact correlated with reduced infectability.

**Table 1 vaccines-11-00509-t001:** Expression of molecular determinants related to viral entry in COVID-19 in children and adults.

Molecule	Expression in Children versus Adults
ACE2	Lower in children than in adults [66,67], Higher in children than in adults [68] higher in children than older (>50 years) adults [69]
TMPRSS2	Lower in children than in adults [66,70]
ADAM17 (TACE)	Lower in children than in adults [71]

### 2.4. Cytokine Profiles and Innate Immunity in Children versus Adults

The clinical condition of adults with COVID-19 depends on cytokine activity and, therefore, understanding cytokine dysregulation is essential in improving clinical outcomes. However, in children, there seems to be no clear connection between cytokine reactivity and disease severity. In fact, cytokines may contribute to immune protection in children as suggested in the previous paragraph. Table 2 shows that Il-17A levels are elevated in COVID-19-infected infants compared to adults. IL-17A is a proinflammatory cytokine produced by different cell types such as natural-killer cells (NK cells), CD4+ T cells, and CD8+ T cells. While IL-17A-induced inflammation in adults is correlated with worse disease outcomes, this has not been observed in the pediatric population [72], and this study suggests that IL-17A or IL-17A producing cells may play a protective role in children. The functional status of NK cells in adults, which deteriorates as part of the aging process and following COVID-19 infectivity, could contribute to this difference [73,74]. Furthermore, effector cytokines such as IFNα and TNFα, both produced by NK cells among others, are decreased in adults (Table 2). In support of the foregoing, NK cell levels increase with age and adults with COVID-19 exhibit lower absolute NK cell counts compared to their healthy counterparts. 

Another age-associated change in innate immunity is impaired pattern recognition receptor (PRR) signaling. Studies have shown that SARS-CoV-2 molecules activate the innate immune system through toll-like receptors (TLRs) [75,76]. TLR-induced cytokine production is decreased in elders, both originated from monocyte-derived dendritic cells (MoDcs) and plasmacytoid dendritic cells (pDcs) [77]. Conversely, cytokine storm is associated with overactivation of TLR pathways in adults. It is unclear whether this overwhelming response also promotes the development of MIS-C in children. However, aging is characterized by remodeling of the immune system, resulting in a higher pro-inflammatory tonus, which could contribute to the excessive TLR response in adults. The importance of a functional balance in TLR activity also relates to the production of interferons (IFNs). IFN production by TLR activation is important to impede virus replication. Type I and II IFN responses in the upper airways are more robust in pediatric than in adult patients, while this is shown to be the opposite for systemic IFN responses [78,79]. Vigorous local IFN responses may account for creating a strong barrier against systemic viral entry and replication in the pediatric population. On the contrary, systemically imbalanced and delayed IFN responses in adults may further contribute to disease progression [80,81].

### 2.5. Adaptive Immunity in Children versus Adults 

Evidence indicates that T cells play a major role in the elimination of SARS-CoV-2 infection [82]. In our previous work, we explained functional differences in CD8+ T cell responses between the young and older population. For example, thymic involution in aging results in shrinkage of the naïve CD8+ T cell compartment and reduced T cell receptor (TCR) diversity [22]. Furthermore, continuous antigenic stimulation can lead to T cell exhaustion, resulting in impaired intracellular TCR signaling. Molecular markers on T cells potentially involved in exhaustion, such as Tim-3 and PD-1, have been linked to disease progression [83]. Considering the foregoing, recent work shows that pediatric patients react with increased magnitude of SARS-CoV-2-specific CD4+ and CD8+ T cell responses compared to adults [84]. It is noteworthy that opposite findings have also been reported [85,86], which could be due to small sample sizes and variance within groups. Even if adults and children are capable of mounting equivalent SARS-CoV-2-specific T cell responses, distinctions in IFN signaling in adults could negatively affect the cytokine response, leading to impaired T cell activity [87]. In addition to differences in T cell responses between young and adult patients, lower T cell frequencies have been observed in older (>60 years) patients [83], but data showing no significant differences in absolute T cell counts between age groups have also been reported [88]. Reduction in total T cell counts in adults is associated with poor outcomes [83]. In support of the latter, it has been demonstrated that T cells from COVID-19-infected patients are more likely to undergo apoptosis and contribute to lymphopenia [89]. Interestingly, a study observed mitochondria-mediated T cell apoptosis, induced by elevated levels of IL-6 and TNFα, in adult but not in child COVID-19 patients [90]. Again, underlying age-dependent differences in the immune system such as increased mitochondrial accumulation and reduced autophagy have been suggested to play a part in this difference [90]. Interestingly, some of the cytokines reported to be induced less in early life in response to SARS-CoV-2 infection, including IL-2, IL-10, and TNFα, were also correlated with more severe COVID-19 disease in adult patients [91]. An interesting concept that was postulated recently is the idea that energy allocation trade-offs between immune responses and child growth likely favor growth unless SARS-CoV-2 represents a serious threat to survival. The resulting decision to mount an inflammatory response to a SARS-CoV-2, or conversely, to tolerate it, could be different in a growing child as compared with an adult [92]. Although this is a plausible concept, it would prove challenging to demonstrate evidence towards this concept, especially because children and adults often experienced different levels of sequestering and social distancing/masking during the pandemic, potentially creating a bias in exposure to strains with variable levels of virulence, as well as exposure to other pathogens such as RSV. 

The role of B cell immunity in COVID-19 remains a matter of debate. Evidence in the literature regarding antibody titers, antibody classes, and isotypes is conflicting. Weisberg et al. (2021) observed that infants mainly produced IgG antibodies against the S protein while adults exhibited a more expanded breadth of anti-SARS-CoV-2-specific antibodies [93]. They also found an overall lower neutralizing antibody response in the pediatric population while opposite findings have also been indicated [94]. Differences in age group range between these studies could explain these opposite observations. Another study demonstrated similar antibody responses against viral proteins between children and adults [84]. Interestingly, a recent cohort study published in JAMA Network Open found that only a small percentage of COVID-19-infected children produced antibodies against SARS-CoV-2 compared to the adult cohort, despite similar viral loads [95]. It can be postulated that the increased inflammatory state in adults, together with impaired features in T cell immunity, would stimulate the humoral response. In addition, the immune response in children may be driven by innate immunity explaining low antibody titers in this population. In contrast, a new study published in the same scientific journal demonstrated significantly higher spike receptor-binding domain (S-RBD) IgG antibody concentrations in especially young children (younger than 3 years) than adults at different time intervals despite weak cross-reactivity to other human coronaviruses (hCov) [96]. These antibody titers remained detectable up to 1 year after infection. The results of this study align with a large prospective multi-center study on the quality and durability of the humoral response between children and adults against SARS-CoV-2 [97]. Preexisting exposure to seasonal hCoV may contribute to a cross-reactive antibody response in children resulting in higher antibody levels [98]. However, limited pre-existing cross-reactive antibodies against hCoV in healthy children and adults has been observed [99,100]. Understanding the role of pre-existing immunity in COVID-19 may provide important insights into pediatric vaccination strategies and should be further researched.

**Table 2 vaccines-11-00509-t002:** Expression of immunological determinants related to disease outcome in COVID-19 in children and adults.

Molecule	Adults/Children with COVID-19	Possible Impact on Outcome (Negative or Positive)
Il-6	Higher in adults than in children [101,102,103,104,105]	None in children [106,107]/negative in adults [108,109,110]
Il-8	Unknown	Negative in adults [110]
Il-10	Higher in adults than in children [101] No differences [102]	Negative in adults [109,111]
Il-2	Higher in adults than in children [102]	None in children [107]
Il-7	?	Negative in adults [111]
Il-5	Higher in adults than in children [101]	
TNFα	Higher in children than in adults [86] No differences [102,105]	None in children [106,107]
TGFβ	?	Negative in adults [112]
IL-17A	Higher in children than in adults [86,113]	Negative in adults [114] None in children [72]
IFNγ	Higher in children than in adults [86] No differences [102,105]	Negative in adults [115] None in children [107]
NF-kb	?	Negative in adults [116]
Il-12	Higher in children than in adults [101]	
IL-1β	Higher in children than in adults [101]	
CD25^+^ (on CD4^+^ cells)	Higher expression in adults than in children [86]	

## 3. Immune Response to Immunization

Currently available COVID-19 vaccines for adults in the United States are based on mRNA (Pfizer-BioNTech and Moderna), subunit (Novavax), and viral vector (J&J/Janssen) technology. Worldwide, vaccines have been made available for pediatric use (16 or younger) in most countries, although the age groups for which eligibility is defined can vary greatly [117]. To date, the Pfizer-BioNTech COVID-19 vaccine (BNT162b2) is the only vaccine approved by the Food and Drug Administration (FDA) for individuals aged 12–17 years in the U.S. In this age group, vaccine efficacy was shown to be similar as in adults, with only mild/moderate reactogenicity and no serious adverse events [118]. Furthermore, in June 2022, the FDA granted an emergency use authorization for BNT162b2 and the Moderna vaccine in individuals aged 6 months to 5 years [119]. While the ingredients in these pediatric vaccines do not differ from the adult formulations, age-specific dose adjustments have been made. Studies in children aged 6 years and older have labeled BNT162b2 as safe and highly efficacious with neutralizing titers and spike-specific IgG responses similar to adults [118,120,121]. However, it is worth noticing that these trials were conducted during the delta wave. Vaccine efficacy studies in 5-to-11-year-olds and adults during the omicron surge have demonstrated a decline in vaccine protection [122], potentially caused by waning immunity [94,123,124]. This is not entirely surprising, since the omicron strain consists of multiple mutations in the RBD that could explain immune evasion [125]. Interestingly, however, this decrease in efficacy appears to be faster in 5–11-year-olds compared to adolescents [126,127]. 

Age-dependent differences in immune responses after vaccination could contribute to these distinct observations. Considering future pediatric COVID-19 vaccine development, it is essential to understand the differences in immunological mechanisms following vaccination between adults and children. However, the literature on innate and adaptive responses to BNT162b2 is less prevalent in children as compared to adults, although this is a rapidly developing field of study [128,129,130]. In adults, BNT162b2 vaccination has proven to stimulate the innate immune response mainly characterized by circulating IFNγ, which enhances after the secondary dose [131]. Furthermore, vaccination induces spike-specific CD4+ and CD8+ T cell and neutralizing antibody responses. Consistent with these observations, murine models have shown similar immune responses to BNT162b2 vaccination [132]. In these models, NK-cells and CD8+ T cells were perceived as the drivers behind IFNγ production. 

As described earlier, age-dependent molecular distinctions in immune response against COVID-19 likely contribute to the milder course of infection in children. Research should reveal whether COVID-19 vaccination in children induces different immune responses compared to their adult counterparts to further optimize vaccine efficacy. By extension, the immunostimulatory properties of the mRNA or the lipid nanocarrier that can provoke the innate immune system may differ between age groups. Insight into the understanding of the adjuvant mechanisms of mRNA vaccine formulations could further contribute to COVID-19 vaccine development and may ultimately plead for more tailored COVID-19 vaccines as opposed to a one size fits all approach.

## 4. Concluding Remarks and Future Directions

In this article we aimed to integrate our understanding of age-dependent immunological differences which could contribute to future pediatric vaccine development against COVID-19. An important limitation of this review is that the field of pediatric vaccine development for SARS-CoV-2 develops very fast. Therefore, we have tried to emphasize those studies and observations that are thoroughly supported by robust study design and confirmation by other studies, to prevent our review from becoming outdated. It is evident that adaptive immune responses significantly change as age increases. Together with factors affecting innate immunity such as NK cell senescence and functionally distinct PRR responses as described previously [16,17,23,37,133,134,135,136,137], generating similar immune responses in adults and children seems challenging. Maturation of the innate and adaptive immune system in adults may partly explain the more favorable prognosis in children after an infection with COVID-19. While the course of COVID-19 infection differs between adults and children, the composition of the BNT162b2 vaccine, which to date is the only COVID-19 vaccine registered for both adults and children aged 12–17 years, is the same. This may imply that the molecular mechanisms underlying immune protection following immunization in children and adults are equal. However, based on our understanding of the distinct immunological features that protect children from severe disease, this notion could be challenged. Therefore, it is essential to conduct research on the mechanisms of the immune response in children following COVID-19 vaccination. Furthermore, mechanistic studies could provide insights into the involvement of humoral immunity in SARS-CoV-2 clearance. Data regarding the contribution of B cell immunity in response to COVID-19 infection across age groups are scattered thus far. Tailor-made vaccine formulations that take these factors into account may offer better protection, which becomes more important when mutant variants may fuel a new surge of COVID-19.

## Figures and Tables

**Figure 1 vaccines-11-00509-f001:**
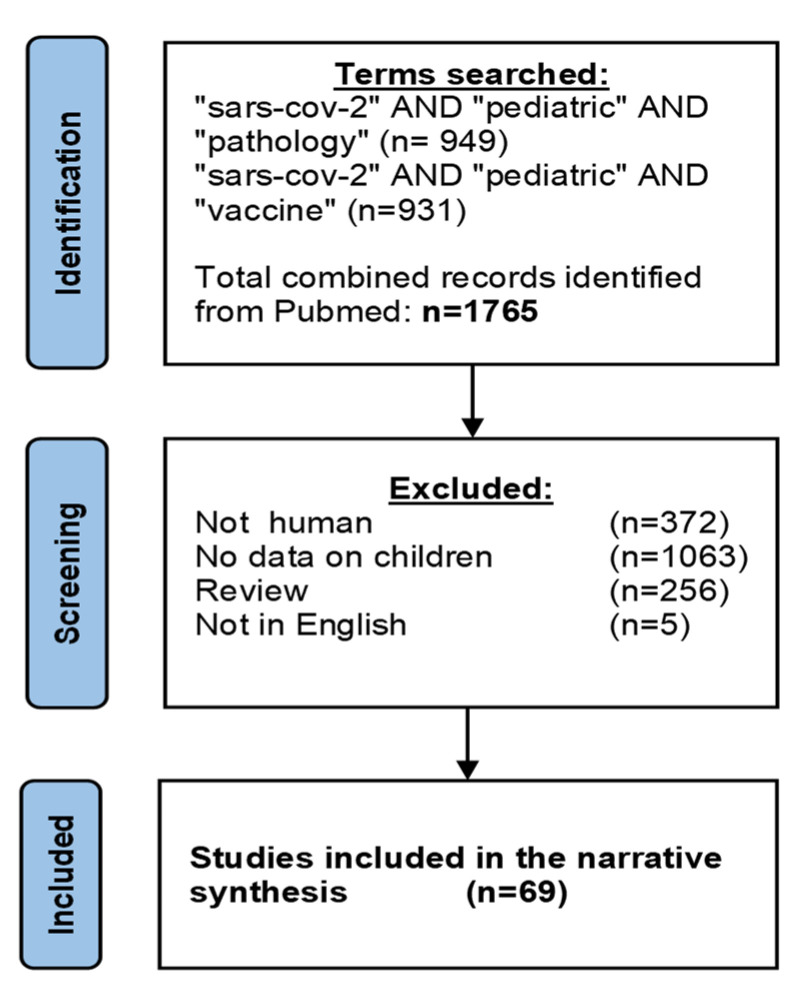
A PRISMA diagram illustrating the criteria for inclusion of original research publications that were included in the narrative synthesis of this review.

**Figure 2 vaccines-11-00509-f002:**
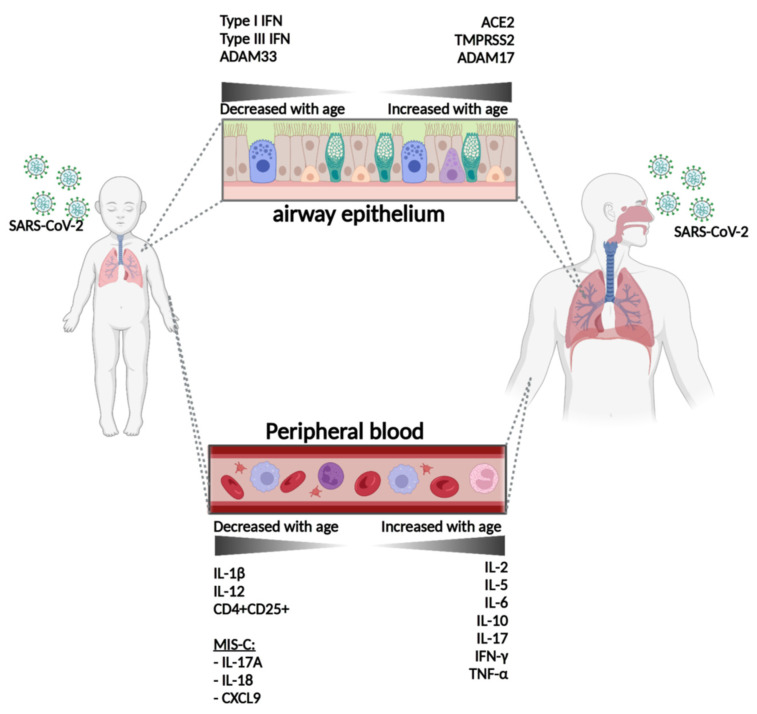
Expression of molecular and immunological determinants related to viral entry and disease outcome in COVID-19 in children and adults. MIS-C: Multisystem Inflammatory Syndrome in Children. Airway epithelium was depicted containing ciliated cells (brown), goblet cell (green), basal cells (pink), and club cells (blue). Image was created with Biorender.com (accessed on 14 February 2023).

## Data Availability

Not applicabl.

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
