# Peer review of "Molecular Determinants of the Early Life Immune Response to COVID-19 Infection and Immunization"

_vaccines, 2023, doi:10.3390/vaccines11030509_

Round 1

Reviewer 1 Report

In their narrative review "Molecular determinants of the early-life immune response to COVID-19 infection and immunization", the authors Elisabeth MS Beijnen , Oludare A Odumade and Simon D. van Haren aim "to integrate our understanding of age-dependent immunological differences which could contribute to future pediatric vaccine development against COVID-19". The text reads well. However, during first reading its analysis and conclusions appear somewhat superficial and figure 1 and table 2 present a list of abbreviations which are not self-explanatory (e.g. the meaning of "COVID-19 levels" and the row below in table 2 or the definition of "MIS-C" in figure 1). 

Thus, the paper unfortunately falls short of the reader's expectations. The paper lacks a defined model/hypothesis to back up its conclusions. Text, tables and figures of the manuscript do not give the information necessary to appreciate the authors' efforts regarding their extensive literature search (Did they adopt a pre-defined search strategy to identify "our understanding"? How were the papers included in the review selected/chosen?). The information given does not allow to understand potential "immunological differences" (between infants', children's and adults' immune responses), and does not focus on the contribution of such differences "to future pediatric vaccine development against COVID-19".

The following points may be used to increase the usefulness of the review for an international audience not familiar with actual concepts in pediatric immunology:

1. Introduction: please add information on important surrogate markers of age-specific immunity to infection in healthy infants and children compared to adults? Which of these mechanisms or molecules are relevant for age-related differences in immune responses to vaccination? Is there any role of the concept of a "layered immune system" [see e.g. Rudd BD Ann Rev Immunol 2020, 38:229] for vaccine-induced memory?

2. Methods: Please add a chapter on search strategy and search results, if possible including something like a PRISMA diagram [see https://prisma-statement.org/prismastatement/flowdiagram.aspx?AspxAutoDetectCookieSupport=1].

3. Results should be organised around 2 main questions [see e.g. Chou JC et al., Nat Immunol 2022; 23:177]:

(1) What are the characteristics of immune responses (mechanisms and/or molecules in the context of mucosal barrier functions, innate and adaptive immunity) in infants, children and adults to infection with SARS-CoV-2?

(2) What roles do these mechanisms and/or molecules play in infants', children's, and adults' immune responses to vaccination against SARS-CoV-2?

4. Discussion: The topic the authors want to cover necessarily reflects a very dynamic field of ongoing scientific research. Interpretation of current literature requires thorough data analysis also regarding differences in laboratory methods used in different studies - the paper completely lacks such information. In addition, one main problem with the interpretation of findings in human beings is the occurrence of different variants of the virus. Is it possible to discern effects of protection against different SARS-CoV-2 variants through previous infection or vaccination? What is the authors' opinion regarding P. Brodin's perspective on "energy allocation" as an explanation of different immune responses to SARS-CoV-2 seen in children and adults [see Brodin P. Immunity 2022; 5:201]?

General remark: given the fact that the publication should be read by an international audience, it would be helpful to explain some abbreviations specific for the U.S. (e.g. AAP = "American Academy of Pediatrics"?); furthermore, the list of available vaccines and their regulatory status should not be limited to approval by the FDA but should also give information on (a) approval and (b) emergency use authorisation by regulatory bodies e.g. in Great Britain, China, the European Union and Brazil for their use in different age groups (best add a table showing this information at a given date).

Reviewer 2 Report

The subject is somewhat clear, and it has been explored much more than the current introduction gives credit. The article presents a good idea.  Although the initial question is interesting, I have a few issues with the study.

The authors should add a section about their motivation to write this review. What is the gap that you aim to address with your review?

1. Title: good

2. Abstract: it captures the appropriate essence of the manuscript. Excellent.

3. Introduction: The introduction identifies the problem that is being addressed in the manuscript and develops and states the purpose of the manuscript.

4. Tables and figures: Quality of figures is so important too. Please provide some high-resolution figures. Some figures have a poor resolution.

5. References: I have verified all references and all key references are correct.

7. Discussion:

* The authors don't discuss the  limitations of the study. It is very important to add this section.

8. Conclusion: The conclusion is justified by the methods and results.

* I have enjoyed reading, and I am in favor of publication after suitable.

Reviewer 3 Report

The manuscript by Beijnen al. reviewed the literature to summarize the differences between children's and adults’ immune responses to SARS-CoV-2 infection and COVID-19 vaccines. This is a timely and well-written review, so I have only a few minor comments.

Lines 131-132: the statement saying that lower expression of ACE2 and TMPRSS2 protects infants from omicron seems contradictory as (1) in line 74 it says that the omicron variant favors the endosomal pathway over TMPRSS2-mediated entry and (2) the number of infected children largely increased during omicron circulation. I would suggest additional clarification of this statement.

Suggestion to remove ‘-‘ in ‘TNF-‘ and ‘IFN-‘ throughout the text as it seems unnecessary and could be confusing, as in line 135

Line 177: the sentence is confusing, perhaps the word ‘-mediated’ should be removed?

Line 227: “significant’ should be ‘significantly’

Figure 1: please add the description to the figure legend of the cells visible in the images; also, the light grey color for human figures does not appear when printed

Table 2: it is not clear what the right column labeled “COVID-19 levels” shows. Please add the description to the text.

Section 3. Was the COVID-19 vaccine approved for ages 6-11? Please add a clarifying sentence.

Lines 272-274: misaligned text indent

Line 267: “However, literature on innate and adaptive responses to BNT162b2 in adults are scarce and absent in children.” The use of the adjectives “scarce” and “absent” is questionable. There are multiple studies analyzing immune responses to both BNT162b2 and Moderna vaccines in adults (e.g. Goel et al., Science 2021; Doria-Rose et al, NEJM 2021; Mantus et al., Cell Rep Med. 2022; Suthar et al., Med 2022, Brasu et al., Nature immunology 2022 etc) but no references were provided. There are also references for children (e.g. Walter et al., N Engl J Med 2022; Bartsch et al., Nature 2022).

Round 2

Reviewer 1 Report

Thank you very much for extensive additional work on the manuscript!